# Stage II Pancreatic Adenocarcinoma after Endovascular Repair of Abdominal Aortic Aneurysm: A Case Report and Literature Review

**DOI:** 10.3390/jcm12020443

**Published:** 2023-01-05

**Authors:** Zihuan Zhang, Duo Li, Tianxiao Wang, Heyuan Niu, Wenquan Niu, Zhiying Yang

**Affiliations:** 1Chinese Academy of Medical Science & Peking Union Medical College, Beijing 100006, China; 2Department of General Surgery, China-Japan Friendship Hospital, Beijing 100029, China; 3Graduate School, Peking University Health Science Center, Beijing 100191, China; 4Department of Hepatobiliary Pancreato-Splenic Surgery, The Affiliated Hospital of Inner Mongolia Medical University, Hohhot 010000, China; 5Institute of Clinical Medical Sciences, China-Japan Friendship Hospital, Beijing 100029, China

**Keywords:** pancreatic adenocarcinoma, abdominal aortic aneurysm, endovascular repair, pancreaticoduodenectomy

## Abstract

Backgrounds: Concomitant abdominal aortic aneurysms (AAA) and gastrointestinal malignancies are uncommon. Endovascular repair (EVAR) is widely used to treat AAA. However, no consensus exists on the optimal strategy for treating AAA when associated with pancreatic adenocarcinoma. In addition, only few reports of pancreaticoduodenectomy (PD) after EVAR exist. Presentation of case: A pancreatic tumor was detected during follow-up after EVAR for AAA in an 83-year-old female patient. The diagnosis was high-grade intraepithelial neoplasia. Modified pylorus-preserving pancreaticoduodenectomy was safely performed. The patient recovered moderately and was discharged two weeks after surgery. The pathological diagnosis was middle-grade pancreatic ductal adenocarcinoma. The patient survived for 24 months with no recurrence or cardiovascular complications. Conclusions: Conducting periodic follow-ups after AAA surgery is helpful for the early discovery of gastrointestinal tumors. EVAR surgery is safe and feasible and thus recommended for AAA patients with pancreatic cancer, although it may increase the risk of cancer. The stage of malignancy and post-EVAR medical history can be valuable in evaluating the benefits of pancreatic surgery for such cases.

## 1. Introduction

Pancreatic cancer is one of the most malignant gastrointestinal tumors with insidious onset and rapid progression. This cancer is characterized by high aggressiveness, metastasis, and recurrence rate. According to the latest data from the National Cancer Institute, the five-year survival rate of pancreatic cancer is only 11.5% [1]; only 20% of cases are resectable at the time of discovery [2].

Case reports of abdominal aortic aneurysms (AAAs) complicated with pancreatic cancer are rare in the medical literature. Previously, surgical resection was the recommended treatment for abdominal aorta. However, a simultaneous resection of AAA and tumors of the digestive system can enormously increase the trauma and risks of artificial vessel infection. In addition, staged resection of AAA and tumor lesions can result in adhesion in the area of the first operation, leading to postoperative complications that increase the difficulty of the second operation. This problem is very significant in the case of pancreatic lesions, considering the anatomy of the pancreas and how close it is to the aorta.

With the widespread use of endovascular repair (EVAR), perioperative mortality is significantly reduced, and postoperative recovery time is shortened accordingly [3,4]. In 2006, Sheen et al. reported the first case of an AAA patient treated with pancreaticoduodenectomy (PD) nine days after EVAR [5]. Based on CT scans, the patient had no anastomotic fistula or stent displacement one month after pancreatic surgery. However, a long-term follow-up was not performed and no information regarding further progression of the patient were available [5]. In 2013, Masahiko et al. [6] investigated a patient with AAA who underwent EVAR and reported a pancreatic mass two years after EVAR. Thus, a laparoscopic pancreaticoduodenectomy (LPD) was carried. The mass was pathologically diagnosed as intraductal papillary mucinous adenoma with small foci of carcinoma in situ. This was the first report on the combined treatment of EVAR and LPD. Although the patient developed grade B pancreatic fistula after surgery, it was cured after drug treatment and no tumor recurrence or cardiovascular complications were observed for 18 months. Here, we report a case of PD after endovascular repair for AAA.

### Presentation of Case

An 83-year-old female with a medical history of hypertension, hyperlipemia, and diabetes mellitus was referred to our hospital due to a pancreatic lesion identified as a solid tumor of the pancreas head by contrast-enhanced computed tomography (Figure 1). Her height, weight, and body mass index (BMI) were 172 cm, 65.3 kg, and 22.1 kg/m^2^, respectively. She had undergone EVAR due to AAA at the age of 73 years old. After that, she had regular follow-up visits. The AAA was located on the infrarenal aorta (Figure 1c).

The CT scan showed a solid tumor of 31 mm diameter in the pancreas head, with dilatation of the main pancreatic duct (Figure 1b). Laboratory tests revealed elevated levels of CA199 (>1000 U/mL, normal <27 U/mL) and CA125 (>43.87 U/mL, normal <35 U/mL). Endoscopic ultrasonography-guided fine needle aspiration (EUS-FNA) indicated that a few highly dysplastic glands were seen in the inflammatory exudate without definite interstitial infiltration.

On 2 November 2020, modified PPPD was successfully performed under general anesthesia. During the operation, AAA was found to be located behind the horizontal part of the duodenum. The complex mass of the uncinate process of the pancreas was palpable (approximately 3 × 4 cm in size). The left and right hepatic arteries had normal blood flow and originated from the proper hepatic artery without vascular variation. Modified PPPD procedure was then performed along with regional lymph node dissection (0/12), including no. 8a, 8p, 12a, 12p, 12b, 12c, 13a, and 13b. An intraoperative histological analysis revealed a negative pancreatic transection margin (R0). The patient recovered uneventfully without complications and was discharged from the hospital 12 days after surgery. Postoperative pathological analysis reported moderately differentiated ductal adenocarcinoma of the pancreas (T2N0M0), 3.5 × 3.3 × 3.3 cm in size. No regional lymph node metastases or perineural infiltrations were observed. The patient survived for 24 months with no recurrence or cardiovascular complications.

## 2. Discussion

It is controversial to simultaneously treat concomitant AAA and malignancies in the adjacent region, especially pancreatic tumors. We systematically searched online literature databases (PubMed, Wanfang, and CNKI) for articles containing “gastrointestinal tumor”, “pancreatic cancer”, and “abdominal aortic aneurysm.” We identified a total of 20 cases of pancreatic lesions with AAA and summarized the patients’ age, gender, aneurysm size, treatment method, interval time, pathology, tumor stage, and prognosis (Table 1) [5,6,7,8,9,10,11,12,13,14,15,16]. Fifteen cases were males, one was female and four were undefined. Such ratio is consistent with previous reports identifying the gender as a risk factor for AAA. Mean age of patients was 67.8 y. Patients included 16 cases of malignant pancreatic tumors, three benign tumors and one with chronic pancreatitis. Additionally, an analysis of stage 4 pancreatic cancer patients (6 cases) showed that cases provided with two types of treatments had a better prognosis.

It is widely recognized that risk profiles of AAA and pancreatic neoplasia are similar, which may influence the prevalence and workup of pancreatic tumor in patients with AAA [16]. Patients diagnosed with AAA need to be continuously monitored for aneurysms before and after surgical repair; thus, a computed tomography (CT) scan can help detect gastrointestinal tumor lesions at early stages [17].

As the number of middle-aged and elderly residents increases, the incidence of AAA and pancreatic cancer is consistently on the rise. Interestingly, risk factors for AAA and pancreatic cancer overlap, such as smoking history, prevalence in males, and advanced age [18,19]. Therefore, pancreatic cancer complicated with AAA is not uncommon. However, the surgical strategies for such cases are very challenging for both general and vascular surgeons due to the surgical sequence, interval and expectation [20,21,22]. Many reports on AAA complicated with the gastrointestinal already exist; however, reports of pancreatic cancer are limited [5,6,23]. Abdullah et al. [24] supported the necessity of simultaneous treatment of malignant tumors and AAA in life-threatening cases. EVAR (when possible) followed by resection of malignant lesions were adopted by the majority of centers.

Previously, researchers believed that when pancreatic cancer and AAA co-occurred, PD should be performed first followed by AAA resection. Deiparine et al. [10] reported two cases of pancreatic lesions complicated with AAA (one with pancreatic cancer and one with chronic pancreatitis) achieving good results, suggesting that carrying pancreatic surgery first could provide definite pathological results. The evaluation of cell type and malignant tumor stage would help determining whether the aneurysm should be repaired. It is easier to repair AAA after PD through a retroperitoneal approach than a peritoneal one. The posterior peritoneal approach avoids the destruction associated with PD or Roux-en-Y reconstruction [10]. However, with the development of interventional technologies, EVAR has gradually become the preferable choice for AAA treatment. Compared to AAA resection, EVAR surgery significantly reduces perioperative mortality and shortens the time between the two operations. Because it is minimally invasive, it reduces the interference with other tumors’ treatment [3,4]. Subsequently, researchers have used EVAR treatment followed by pancreatic surgery [5,6]. Because the lesions of the abdominal aorta and pancreas are close to each other, operative field interferes significantly and the risk of concurrent operation is high. Deiparine et al. [10] advocated retroperitoneal incision: PD through the right surgical approach, and left abdominal aortic bypass. EVAR has not been associated with such concerns. However, concerns regarding the elevated mortality rate from EVAR have been raised, which may be attributed in part to an increased cancer burden at a later stage after exposure to external radiation from surgery or monitoring the stent with CT [25]. It may explain the phenomenon mentioned by Portelli et al., that incidental pancreatic tumors are common during the monitoring of aortic aneurysms [16]. Moreover, previous trials have shown a higher rate of local vascular or device-related complications and 30-day reintervention after EVAR compared to surgical resection [26,27,28]. Overall, we believe that EVAR remains the first option for AAA associated with pancreatic cancer, given its ability to reduce early mortality and shorten surgical intervals. For the severity of the postoperative complications of AAA, the open repair of AAA surgery needs artificial blood vessels. It is highly invasive to perform AAA resection and radical resection of digestive system tumors simultaneously due to the increased risk of artificial blood vessel infection [29]. If performed in stages, adhesions in the area of the first surgery may increase the difficulty in the second surgery and the possibility of postoperative complications. In particular, the pancreas is anatomically close to the aorta. Evaluating the likelihood of postoperative pancreatic fistula and early intervention may be beneficial. Regarding the choice of pancreatic surgery methods, Felix et al. [30] conducted a systematic review to compare LPD and PD and pointed out that LPD was safe and feasible, yet obtained apparent advantages compared to PD [30,31,32]. In the previous case report, Masahiko et al. [6] believed that residual abdominal aortic aneurysm had little interference with the operation. Laparoscopic surgery could magnify the surgical field, and so LPD was thought be to an indication for patients with AAA and benign pancreatic tumors [6]. However, due to the long history of AAA and the large aneurysm, we believe that the choice of open surgery is safer and more feasible. Out of consideration of AAA location, we adopted a modified pylorus-preserving pancreaticoduodenectomy (PPPD) (Figure 2), a surgical approach that preserves the horizontal part of the duodenum and the upper part of the jejunum yet reduces the impact of PD on AAA, while ensuring clean surgical margins. This has separated us from other medical institutions. 

With the growing research interest in pancreatic cancer, more and more surgeons are realizing the importance of the meso-pancreas [33,34,35,36], which refers to the complete resection of perineural and lymphatic tissues, in addition to structures surrounding the pancreatic head/uncinate. This approach may decrease the recurrence rate and improve survival due to the complete clearance of the region [37]. However, we did not perform middle pancreatectomy and lymph node dissection in this area, because the location of the aortic aneurysm was adjacent to the middle pancreas, which increased the risk of surgery.

Previous literature has reported that the circumferential resection margin (CRM) consists of the ventral and dorsal pancreatic surfaces, as well as the medial pancreatic margin [38,39]. Adding CRM status to the standard pathological assessment can improve the accuracy of marginal excision [38] and correlate with median survival [37]. Unfortunately, such an approach is not adopted by the pathology department of our hospital.

The effects of chemotherapy on aneurysms should also be considered. Russwurm et al. [12] reported a case of AAA and hepatic metastasis in the pancreas. who received chemotherapy (gemcitabine) as a treatment. During the first four weeks, the patient was given prednisone (10 mg/d). When monitoring the size of AAA, an aneurysm expansion was observed, but when the chemotherapeutic agents were withdrawn, the expansion was stagnated. They hence speculated that the use of chemotherapeutic agents might accelerate the growth of aneurysm [12]. Recently, Maxwell et al. [40] examined the effects of malignant tumors and chemotherapy on AAA and found that aneurysms behaved similarly to those in patients without malignancy. In this case, the patient was older and had poor tolerance to chemotherapy, with a history of cardiovascular and cerebrovascular diseases. Since chemotherapy may increase the risk for cardiovascular, cerebrovascular embolization, and stent embolization, it was not adopted as treatment plan. Blood pressure was controlled before and after the surgery. Fortunately, no post-surgical complications occurred and the patient was discharged smoothly. Enhanced CT scans were performed every six months, and no evident tumor recurrence was found. The patient survived for 24 months.

## 3. Conclusions

Regular follow-ups after AAA surgery can help in the early detection of gastrointestinal tumors. Although EVAR may increase the burden of cancer at a later stage (due to exposure to external radiation from surgery or monitoring the stent with CT), it is recommended for patients with complicated AAA pancreatic cancer. It is also necessary to evaluate whether patients can benefit from pancreatic surgery after EVAR by considering the stage of malignant tumor and their medical history. For patients with advanced pancreatic cancer, most patients receiving two types of treatments usually have a better prognosis, and those receiving only one kind of treatment also have a better prognosis than those without treatment. However, some treatments, such as chemotherapy, may accelerate the growth of AAA and thus should be carefully evaluated.

## Figures and Tables

**Figure 1 jcm-12-00443-f001:**
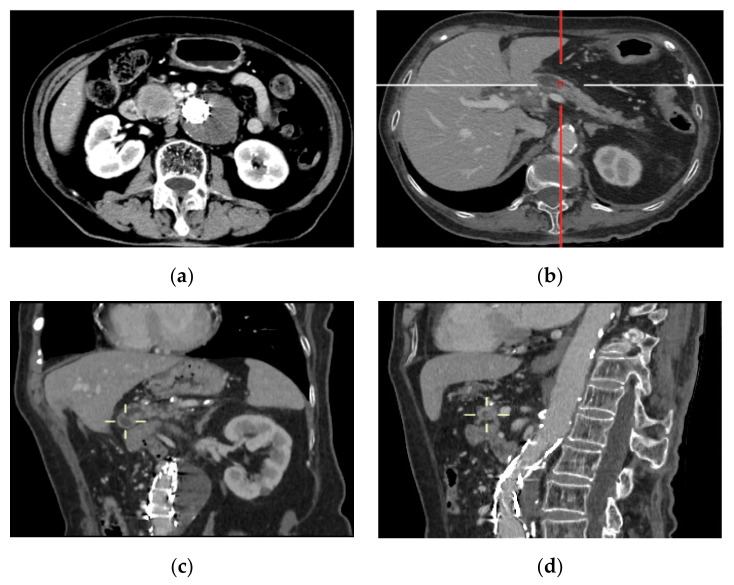
CT images of (**a**) intra-aortic endovascular stent in transverse arterial phase; (**b**) a solid tumor of 31 mm diameter in the pancreas head with dilatation of the central pancreatic duct in transverse venous phase; (**c**) sagittal plane showing an infrarenal AAA and the endovascular stent in coronal arterial phase; (**d**) location of pancreatic cancer and endovascular repair of AAA in sagittal venous phase.

**Figure 2 jcm-12-00443-f002:**
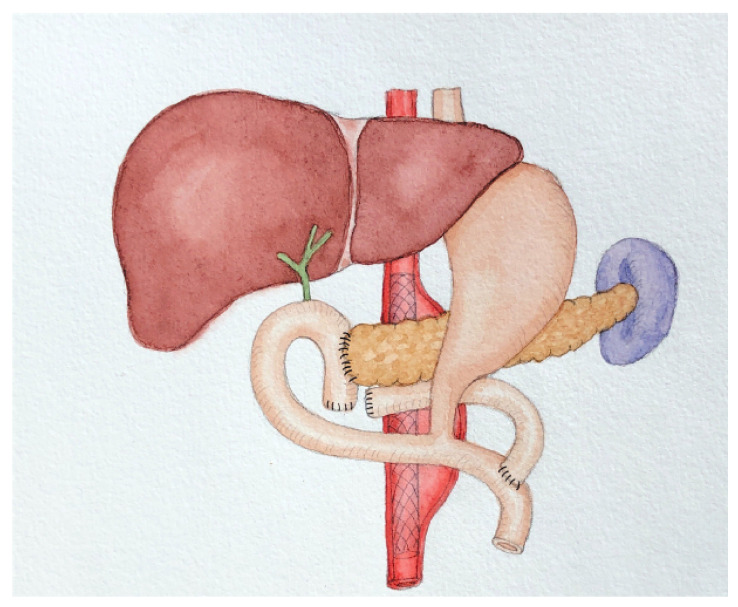
Modified PPPD. PPPD: pylorus-preserving pancreaticoduodenectomy.

**Table 1 jcm-12-00443-t001:** Summary of AAA with pancreatic diseases in the literature.

First Author, Year	Age (Years)	Sex	Size of AAA (cm)	Approach	Interval	Biopsy	Stage	Prognosis (Months)
Szilagy, 1967 [7]	65	M	-	Cholecystojejunostomy	NA	pancreatic adenocarcinoma	4	3
	79	M	-	None	NA	pancreatic adenocarcinoma	4	3
Gibbs, 1973 [8]	48	F	6	One stage	NA	pancreatic cystadenoma	-	15+
Komori, 1993 [9]	-	-	-	Only resection of pancreatic tumor	NA	pancreatic adenocarcinoma	-	-
	-	-	-	Only resection of the aneurysm	NA	pancreatic adenocarcinoma	-	-
	-	-	-	None	NA	pancreatic adenocarcinoma	-	-
Deiparine, 1995 [10]	62	M	9.0	PD first, resection of the aneurysm	2 weeks	Pancreatic islet cell tumor	-	25+
	71	M	6.0	PD first, resection of the aneurysm	2 months	chronic pancreatitis	-	36+
Konno, 1998 [11]	73	M	5.0	AAA first, total PD	2 weeks	NA	4	14
Palm, 2000 [12]	73	M	5.9	Only chemotherapy	NA	NA	4	10
Piecuch, 2003 [13]	73	-	5.5	One stage (Resection of aneurysm + PD)	NA	pancreatic adenocarcinoma	-	-
Ryan, 2005 [14]	58	M	3.8	Chemotherapy +PD	NA	pancreatic adenocarcinoma	4	3
Sheen, 2006 [5]	60	M	6.0	EVAR first, PD	9 days	NA	2	-
Veraldi, 2008 [15]	64	M	5.6	One stage (EVAR + PD)	NA	pancreatic adenocarcinoma	1	28
	73	M	6.2	EVAR first, PD	3 days	pancreatic adenocarcinoma	1	37
	71	M	5.5	One stage (Resection of aneurysm + PD)	NA	pancreatic mucinous cystic adenoma	-	10+
Kawguchi, 2013 [6]	70	M	3.1	LPD	NA	IPMN + small foci of carcinoma in situ	1	18
Tremont, 2021 [16]	62	M	8.6	Chemotherapy	NA	pancreatic adenocarcinoma	4	0.3
	66	M	3.5	None	NA	benign cystadenoma	-	134
	85	M	2.8	Chemotherapy + surgery	NA	pancreatic adenocarcinoma	4	45

Note: AAA: abdominal aortic aneurysm; Interval: Interval between AAA repair and cancer treatment; M: male; F: female; PD: pancreaticoduodenectomy; LPD: Laparoscopic pancreaticoduodenectomy; EVAR: endovascular repair.

## Data Availability

The data presented in this study are available in this article.

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
