# Peer review of "Stage II Pancreatic Adenocarcinoma after Endovascular Repair of Abdominal Aortic Aneurysm: A Case Report and Literature Review"

_jcm, 2023, doi:10.3390/jcm12020443_

Round 1
Reviewer 1 Report
Please state the TNM staging as UICC with number of LN or AJCC with number of LN.
Please state the technique of PD provided. you mention modified. what was modified and different from a standard PD? what is your philosophy on arterial pathologies and pancreatic tail preserving PD? pancreatic anastomotic fistula and sentinel bleeding? please mention complication management in your discussion because in patients with history of EVAR, hepatic stenting or stenting of SMA is more difficult. How was the arterial supply of the Liver? left and right hepatic artery origin?. how did you take care of the posterior dissections in order to secure CRM negativity?
did you harvest the lymphatic tissue around the aorta and SMA? did you preserve the perineural sheet? mesopancreatic excision for pancreatic cancer is the standard in oncology, Does AAA prevent surgeons from performing mesopancreatic excision?
In modern medicine it should not be controversial to treat stable AAA Patients for a resectable malignancy. what is your alternative? please remove this sentence.
Author Response
we are very grateful to you for allowing us to revise our manuscript. Thank you very much for your valuable time and helpful comments. The following are the responses and revisions we have made in response to the reviewers’ questions and suggestions on an item-by-item. Please see the attachment.

Reviewer 2 Report
The article entitled, “Pancreatic adenocarcinoma at stage Ⅱ after endovascular repair for abdominal aortic aneurysm: a case report and literature review” is a case report cum literature review as the title suggests. It is definitely not the first case report of pancreatic cancer after EVAR, which is performed to repair AAA. But it is also clear that every case report, would add to the information about treatment options available to treat future patients. The case is of a women, who had EVAR for her AAA at the age of 73 and was diagnosed with pancreatic cancer 10 years later at the age of 83.
Comment:
1) Include, the articles, “A population-based cohort study examining the risk of abdominal cancer after endovascular abdominal aortic aneurysm repair” (PMID: 30583890) in your literature review section and discuss it as it is a population study about developing abdominal cancers following EVAR. The authors claim EVAR as a safe procedure, with very little risk involved. But could it be a cause to speed up abdominal cancers, like pancreatic cancer? Include other risks about EVAR. This discussion will help the readers get a complete overview of the benefits and risks about EVAR.
2) Cite the review article, “Management of concomitant cancer and abdominal aortic aneurysm” (PMID: 21559270), which discusses similar ideas as discussed in this manuscript.
Author Response
we are very grateful to you for allowing us to revise our manuscript. Thank you very much for your valuable time and helpful comments. The following are the responses and revisions we have made in response to the reviewer's questions and suggestions on an item-by-item. Please see the attachment.

Round 2
Reviewer 1 Report
Dear authors,
CRM are not the resection margins you stated. CRM refers to the posterior medial and central resection margin of the pancreas. Please cite Esposito et al. Most pancreatic resections are R1 resections.
up to date only one surgical paper exists on Mesopancreas and the mesopancreatic excision with pathological evaluation of the Mesopancreas. Please cite Safi et al. Mesopancreatic resection for pancreatic ductal adenocarcinoma improves local disease control and survival.
Author Response
We are very grateful to you for allowing us to revise our manuscript. Thank you very much for your valuable time and comments. The following are the responses and revisions we have made in response to the reviewer’s questions and suggestions on an item-by-item basis. Please see the attachment.
